# SPARSE MATRIX PRODUCTS FOR NEURAL NETWORK COMPRESSION

## ABSTRACT

Over-parameterization of neural networks is a well known issue that comes along with their great performance. Among the many approaches proposed to tackle this problem, low-rank tensor decompositions are largely investigated to compress deep neural networks. Such techniques rely on a low-rank assumption of the layer weight tensors that does not always hold in practice. Following this observation, this paper studies sparsity inducing techniques to build new sparse matrix product layer for high-rate neural networks compression. Specifically, we explore recent advances in sparse optimization to replace each layer's weight matrix, either convolutional or fully connected, by a product of sparse matrices. Our experiments validate that our approach provides a better compression-accuracy trade-off than most popular low-rank-based compression techniques.

## 1 INTRODUCTION

The success of neural networks in the processing of structured data is in part due to their over-parametrization which plays a key role in their ability to learn rich features from the data (Neyshabur et al., 2018). Unfortunately, this also makes most state-of-the-art models so huge that they are expensive to store and impossible to operate on devices with limited resources (memory, computing capacity) or that cannot integrate GPUs (Cheng et al., 2017). This problem has led to a popular line of research for "neural networks compression", which aims at building models with few parameters while preserving their accuracy.

**State of the art techniques for neural network compression.** Popular matrix or tensor decomposition methods including Singular Value Decomposition (SVD), CANDE-COMP/PARAFAC (CP) and Tucker have been used to address the problem of model compression by a low-rank approximation of the neural network's weights after learning. Sainath et al. (2013) describe a method based on SVD to compress weight matrices in fully connected layers. Denton et al. (2014); Lebedev et al. (2015); Kim et al. (2016) generalize this idea to convolutional layers and then reduce the memory footprint of convolution kernels by using higher-order low-rank decompositions such as CP or Tucker decompositions.

Besides, the Tensor-Train (TT) decomposition has been explored to compress both dense and convolutional layers after a pre-training step (Novikov et al., 2015). This approach may achieve extreme compression rates but it also have impractical downsides that we demonstrate now. In a TT format, all the elements of a $M$-order tensor are expressed by a product of $M$ matrices whose dimensions are determined by the TT-ranks $(R_0, R_1, \ldots, R_M)$. For each of the $M$ dimension of the initial tensor, the corresponding matrices can be stacked into an order 3 tensor called a "core" of the decomposition. Hence, the layer weight is decomposed as a set of $M$ cores of small dimensions. Novikov et al. (2015) use this tensor representation to factorize fully connected layers. They first reshape the matrix of weights into an $M$-order tensor, then apply the TT decomposition. By choosing sufficiently small $R_m$ values, this technique allows to obtain a high compression ratio on extremely wide *ad hoc* neural architectures. Garipov et al. (2016) have adapted this idea to convolutional layers. However, the current formulation of such TT convolutional layer involves the multiplication of all input values by a matrix of dimension $1 \times R_1$ thus causing an inflation of $R_1$ times

the size of the input in memory. This makes the available implementation (Garipov, 2020) unusable for recent wide convolutional networks at inference time.

Other compression methods include unstructured pruning techniques that we review more in details in Section 2.3 and structured pruning techniques that reduce the inner hidden dimensions of the network by completely removing neurons (Anwar et al., 2017). According to the recent paper of Liu et al. (2018) however, these techniques are more akin to *Neural Architecture Search* than actual network compression. Finally, quantization-based compression maps the columns of the weight matrices in the network to a subset of reference columns with lower memory footprint (Guo, 2018).

**Sparse matrices product for full rank decompositions.** We are specifically interested in high-rate compression of neural networks via the efficient factorization of the layer weight matrices. Most known approaches to layer decomposition usually makes low-rank assumption on the layer weight tensors which does not always hold in practice. As we will show in the experiments, this makes the Tucker and SVD based techniques unable to effectively reach high compression rates for standard architectures including both convolutional and fully connected layers, such as VGG19 or ResNet50, whose weight matrices usually exhibit full rank. In this paper, we propose instead to express the weight matrices of fully-connected or convolutional layers as a product of sparse factors which contains very little parameters but still can represent high-rank matrices. Moreover, products of matrices with a total sparsity budget are strictly more expressive than single matrices with that sparsity (Dao et al., 2019), which motivates our interest in products of multiple matrices. Usually, a linear operator (a matrix) from $\mathbb{R}^D$ to $\mathbb{R}^D$ has a time and space complexities of $\mathcal{O}(D^2)$. But some well known operators like the Hadamard or the Fourier transforms can be expressed in the form of a product of $\log D$ sparse matrices, each having $\mathcal{O}(D)$ non-zero values (Dao et al., 2019; Magoarou & Gribonval, 2016). These linear operators, called fast-operators, thus have a time and space complexities lowered to $\mathcal{O}(D \log D)$. This interesting feature of fast-operators have inspired the design of new algorithms that learn sparse matrix product representations of existing fast-transforms (Dao et al., 2019) or even that computes sparse product approximations of any matrix in order to accelerate learning and inference (Magoarou & Gribonval, 2016; Giffon et al., 2019). Even though these new methods were initially designed to recover the $\log D$ factors corresponding to a fast-transform, they are more general than that and can actually be used to find a factorization with $Q < \log D$ sparse matrices.

**Contributions.** We introduce a general framework for neural network compression using the factorization of layers into sparse matrix products. We explore the use of the recently proposed `palm4MSA` algorithm (Magoarou & Gribonval, 2016) on every layer of a pre-trained neural network to express them as a product of sparse matrices. The obtained sparse matrices are then refined by gradient descent to best fit the final prediction task. When there is only one sparse matrix in the decomposition, our approach recovers the simple procedure of hard thresholding the weights of a matrix after pre-training. We evaluate the effect of different hyper-parameters on our method and show that layers can be factorized into two or three sparse matrices to obtain high compression rates while preserving good performance, compared to several main state-of-the-art methods for neural network compression.

## 2 Learning sparse matrix products for network compression

We describe how to compress NN weight matrices by sparse matrix factorization. We call our procedure PSM for Product of Sparse Matrices. It is obvious to see that a product of sparse matrices with a given sparsity budget can recover a full rank matrix or a matrix with more non-zero values than the initial sparsity budget. This observation motivates the use of a sparse matrix factorization in place of usual low-rank decomposition and sparsity inducing techniques for neural network compression. We first recall linear transform operations in fully-connected and convolutional layers. Then, inspired by recent work on learning linear operators with fast-transform structures, we propose to use a product of sparse matrices to replace linear transforms in neural networks. We also introduce a procedure to learn such factorization for every layers in deep architecture. Finally, we review some known neural network compression techniques that appear as particular cases of our framework.

## 2.1 Weight matrices as product of sparse matrices

Fully-connected and convolutional layers are based on the computation of linear operations.

In a fully-connected layer, the output $\mathbf{z} \in \mathbb{R}^{D'}$ is simply given by $\mathbf{z} = \mathbf{a}(\mathbf{Wx})$ where $\mathbf{a}$ is some non-linear activation function. $\mathbf{W} \in \mathbb{R}^{D' \times D}$ is the weight matrix of the layer and $\mathbf{x} \in \mathbb{R}^{D}$ is the output of the preceding layer.

The linear operation in a convolutional layer can be represented by a doubly-block Toeplitz matrix (Wang et al., 2020). An other way to perform the operation is to employ reshaping operators to represent the linear operator as a dense matrix applied to all the patches extracted from the input (Garipov et al., 2016). In this work, we focus on this latter representation of the convolution operation. More formally, let $\mathbf{r}_S : \mathbb{R}^{H \times W \times C} \mapsto \mathbb{R}^{HW \times CS^2}$ be the reshape operation that creates the matrix of all vectorized patches of size (height and width) $S^2$ on an input image with $C$ channels. The matrix of $K$ filters $\mathbf{W} \in \mathbb{R}^{CS^2 \times K}$ can then be applied to these patches (multiplied with $\mathbf{r}_S$) to produce the output of the convolutional layer in a matrix shape. Finally, a second reshape operator $\mathbf{t} : \mathbb{R}^{HW \times K} \mapsto \mathbb{R}^{H \times W \times K}$ is applied on the feature map matrix to reconstruct the output tensor of the layer $\mathcal{Z} \in \mathbb{R}^{H \times W \times K}$. Altogether, the convolution operation can be written as $\mathcal{Z} = \mathbf{a}(\mathbf{t}(\mathbf{r}_S(\mathcal{X})\mathbf{W}))$ where $\mathbf{a}$ is some non-linear activation function and $\mathcal{X}$ is the output 3-D tensor of the preceding layer. We preserve simplicity in notation here, assuming without loss of generality that the stride used by $\mathbf{r}_S$ is equal to 1 and that the input tensor is padded with $\lfloor \frac{S}{2} \rfloor$ zeros vertically and horizontally. The whole process is depicted in Supplementary Material A.2.

Our general idea is to replace the weight matrix of each neural network layer with a product of $Q$ sparse matrices, hence reducing the storage and computational complexities of the layer. Indeed, for an initial matrix of dimension $(D \times D')$, if all sparse matrices store $\mathcal{O}(D)$ non-zero values, then the total complexity of the product becomes $\mathcal{O}(QD)$ instead of $\mathcal{O}(DD')$. To define a fast-transform operator, one would use $Q = \log D$ but in practice we show that we can chose much smaller $Q$ and achieve huge compression rates without lowering much the performance. Supplementary Material A.1 illustrates the effect of our compression scheme on a simple architecture including one convolution layer and a single dense layer. Given an input vector $\mathbf{x} \in \mathbb{R}^{D}$, expressing the weight matrix $\mathbf{W} \in \mathbb{R}^{D' \times D}$ of a fully connected layer as a product of sparse matrices gives output $\mathbf{z}$ such that:

$$\mathbf{z} = \mathbf{a}\left(\prod_{i=1}^{Q} \mathbf{S}_i \mathbf{x}\right), \tag{1}$$

where $||S_i||_0 = \mathcal{O}(D)$ so that the complexity in time and space of this layer is reduced to $\mathcal{O}(QD)$ instead of $\mathcal{O}(DD')$.

Similarly, in the convolution layers, the output $\mathcal{Z} \in \mathbb{R}^{H \times W \times K}$ is obtained from an input tensor $\mathcal{X} \in \mathbb{R}^{H \times W \times C}$:

$$\mathcal{Z} = \mathbf{a}\left(\mathbf{t}\left(\mathbf{r}_S(\mathcal{X})\prod_{i=1}^{Q} \mathbf{S}_i\right)\right), \tag{2}$$

where $||S_i||_0 = \mathcal{O}(\max(S^2 C, K))$ so that the time complexity of the layer is reduced from $\mathcal{O}(HWCS^2K)$ to $\mathcal{O}(HWQ \cdot \max(CS^2, K))$ and the complexity in space is reduced from $\mathcal{O}(CS^2K)$ to $\mathcal{O}(Q \cdot \max(CS^2, K))$.

Since there is no constraint on the rank of factors, the sparse matrix products of each layer can reach full rank, unlike low-rank decomposition methods. Moreover, the reconstruction of a sparse matrix product with a total of $\mathcal{O}(QD)$ non-zero values can produce a matrix with more than $\mathcal{O}(QD)$ non-zero values. This is consistent with the intuition that a product of sparse matrices can be more expressive than a single sparse matrix.

## 2.2 Full Neural Network Compression

The full compression pipeline we propose includes first the learning of a standard NN, second the compression of each layer independently as a product of sparse matrices, and finally a fine tuning of the compressed NN whose layers are all expressed as PSM layers.

The second step requires approximating each weight matrix $\mathbf{W}$ (of a dense or a convolutional layer) as a product of sparse factors, which is cast as the following optimization problem:

$$\min_{\{\mathbf{S}_i\}_{i=1}^Q} \left\| \mathbf{W} - \prod_{i=1}^Q \mathbf{S}_i \right\|_F^2 + \sum_{i=1}^Q \delta_{\mathcal{E}_i}(\mathbf{S}_i), \tag{3}$$

where for each $i \in [\![Q]\!]$, $\delta_{\mathcal{E}_i}(\mathbf{S}_i) = 0$ if $\mathbf{S}_i \in \mathcal{E}_i$ and $\delta_{\mathcal{E}_i}(\mathbf{S}_i) = +\infty$ otherwise. $\mathcal{E}_i$ is the set of solutions that respect a sparsity constraint (e.g. number of non zeros values). Although this problem is non-convex, non-differentiable, and the computation of a global optimum cannot be ascertained, the `palm4MSA` algorithm proposed by Magoarou & Gribonval (2016) is able to learn such factorization by finding a local minimum with convergence guarantees. For more details about `palm4MSA`, see Supplmentary Materiel A.4.

Once every layer's weight matrix is approximated by a product of sparse matrices, these PSM layers are assembled in a compressed NN which is refined to optimize the initial task objective while the sparsity support of all factors is kept fixed.

## 2.3 Related work

Some techniques based on inducing sparsity in neural connections, *e.g.* zeroing single weights in layer tensors, can be seen as particular cases of our method. The most straightforward approach to this is to simply remove the weights with lowest magnitude until a given sparsity ratio is reached. This can be done in a very trivial fashion by just removing weights on a pre-trained network and then finetuning the remaining connections. This method can be seen as the particular case of ours when there is only one factor to approximate weight matrices (i.e., $Q = 1$). As we show in the experiments, this method doesn't allow high compression rate without degradation of the accuracy. Zhu & Gupta (2017) proposed instead to intertwine the removal of the connections by finetuning remaining weights, achieving better classification performance. Others, approaches for inducing sparsity in the network were proposed (Molchanov et al., 2017; Louizos et al., 2018-06-22) but they do not seem to offer performance improvements in general settings (Gale et al., 2019).

The idea of replacing layers by sparse factorization has been previously explored but restricted to particular structures. In `Deep Fried Convnets`, Yang et al. (2015) propose to replace dense layers of convolutional neural networks by the `Fastfood` approximation (Le et al., 2013). This approximation is a product of diagonal matrices, a permutation matrix, and a Hadamard matrix which can itself be expressed as a product of $\log D$ sparse matrices (Dao et al., 2019). The `Fastfood` approximation thus provides a product of sparse factors and from this perspective, the `Fastfood` layer proposed by Yang et al. (2015) is a particular, constrained, case of our more general framework. Moreover, the `Deep Fried Convnets` architecture is based on the Hadamard matrix that imposes a strong structural constraint on the factorization, which might not be suitable for all layers of a deep architecture.

The term sparse decomposition used in Liu et al. (2015) for network compression refers to separate products between dense and sparse matrices to represent the weights of the convolution kernels in a network. Finally, Wu et al. (2019) have recently proposed a very similar framework than ours along with a regularization strategy to learn the sparsity in the sparse factors but their method does not allow for more than two sparse factors and the compression of the convolutional layers is not considered although best performing architectures tend to store most of their parameters in these layers.

## 3 Experiments

Section 3.1 details the experimental settings and parameters to ensure reproducibility. We provide an in depth analysis of our method in Section 3.2. Finally we report in Section 3.3 a comparative study of our method with state-of-the-art methods.

### 3.1 Experimental setting

Our analysis is focused on image classification tasks, we investigate the compression of standard architectures (pretrained models) with our approach and with a few state of the art methods. We evaluate all methods by measuring both the compression ratio and the accuracy of compressed models. We first present implementation details and datasets, then we detail the baselines and the hyperparameters we used.

**Implementation details.** The code was written in Python, including the `palm4MSA` algorithm (it will be made available on github). NNs were implemented with `Keras` (Chollet, 2015) and `Tensorflow` (Abadi et al., 2015). Due to the lack of efficient implementation of the sparse matrices in Tensorflow, we had to hijack the implementations of the dense matrix product and convolution offered by Keras in order to make the learning of networks as efficient as possible (See Supplementary material A.3 for details).

**Datasets and Neural Architectures.** Our experiments are conducted on four standard image classification data sets of varying difficulty: `MNIST` (Deng, 2012), `SVHN` (Netzer et al., 2011), `CIFAR10`, `CIFAR100` (Krizhevsky, 2009). Pretrained NNs to be compressed are classically used with these data sets i.e. `Lenet`(LeCun et al., 1998), `VGG19`(Simonyan & Zisserman, 2015), `Resnet50` and `Resnet20` (He et al., 2016). Details on datasets and neural architectures may be found in Supplementary Material A.5.

**Competing baselines.** We present below the baselines and the variants of our approach that we considered. In all cases the methods were applied on the same pre-trained models and all compressed models were fine-tuned after compression:

- `Low-rank factorization` methods, including Tensor-Train decomposition (Novikov et al., 2015; Garipov et al., 2016) (named *TT* hereafter) and an hybrid of Tucker decomposition (Kim et al., 2016) and SVD (Sainath et al., 2013) (named *Tucker-SVD*) where Tucker and SVD decomposition are used respectively for the compression of convolutional and dense layers.

- Two `sparsity inducing pruning techniques`. The first one is a simple magnitude based projection of weight matrices on a sparse support. This can be seen as a particular case of our model where only one sparse factor is required. We name this strategy "Hard pruning" (*HP* in the following). The second method, named *Iterative pruning* (*IP*) is the iterative strategy proposed by Zhu & Gupta (2017), which refine magnitude based projection with finetuning steps.

- Finally, we evaluate the interest of using the `palm4MSA` algorithm to discover a sparse matrix decomposition of the layers compared to some random decomposition. More precisely, we evaluate (I) the performance of a model whose layers would be decomposed by `palm4MSA` but whose weights would be re-initialized while peserving the sparsituy support; and (II) the performance of a model whose weights and sparsity support would be randomly sampled at initialization.

Note that we initially considered *Deep Fried convnets* as an additional baseline but we finally did not include it on our experimental study since this method is originally dedicated to compress fully connected layers and our attempts to make it able to compress convolutional layers as well actually failed, preventing the method to be applied to compress many state of the art architectures that contain mostly convolutional layers.

**Hyper-parameters.** The stopping criteria used for the `palm4MSA` algorithm is 300 iterations or a relative change between two iterations below $10^{-6}$. The projection method is the one used in Magoarou & Gribonval (2016). With $K$ the desired level of sparsity, this method ensures that each sparse factor contains at least $K$ non-zero values per row and per column and at most $2K$ on average.

For experiments with random sparsity support, the initialization of the weights must be adapted to the reduced number of connections in the layer. To do this, we adapt the initializations `Xavier` (Glorot & Bengio, 2010) and `He` (He et al., 2015) to the initialization of sparse matrices (See Supplementary Material A.3).

We chose $M = 4$ cores for the Tensor-Train decomposition of any tensor and the maximum possible rank value of the decompositions is specified by the value $R$ in the experiments.

In the hybrid of Tucker and SVD decomposition, the rank of the Tucker decomposition is automatically detected by the Variational Bayes Matrix Factorization method, as explained by Kim et al. (2016); the rank of the SVD in the dense layers is chosen such that only a certain percentage (specified in the experiments) of the singular values are kept.

Fine-tuning of the `Lenet` network was done with the `RMSProp` optimizer and 100 learning epochs. The `VGG19` and `Resnet` networks are fine-tuned with `Adam` (Kingma & Ba, 2014) optimizer and respectively 300 and 200 epochs. For each compression method and configuration, the best learning rate is chosen using the classification error on a validation sample after 10 iterations, with learning rate values in $\{10^{-3}, 10^{-4}, 10^{-5}\}$. Standard data augmentation is applied: translation, rotation and flipping.

## 3.2 Analysis of the method

We first provide an in-depth analysis of our method to validate the use of `Palm4MSA` for the decomposition of layer weight matrices into a product of sparse matrices. We then evaluate the impact of hyper-parameters $Q$ and $K$ on model accuracy and compression rate.

**Approximation error.** We evaluate the quality of the `Palm4MSA` approximation of original weight matrices, and its influence on the final performance of the method. We report results gained on the compression `VGG19` trained on `CIFAR100` as illustration. Figure 2 shows the approximation error between the product of $Q$ sparse factors $\tilde{\mathbf{W}} := \prod_{q=1}^{Q} \mathbf{S}_q$ and the original weights $\mathbf{W}$ for each layer. It is computed as the normalized distance between the matrices: $error = \|\mathbf{W} - \tilde{\mathbf{W}}\|_F^2 / \|\mathbf{W}\|_F^2$.

Figure 2 further shows that higher $K$, *i.e.* the minimum number of non-zero values per row and per column, yield better approximation. We observe that for some layers, the error can be high, despite relative good accuracy performance observed in Table 1, suggesting the use of the `Palm4MSA` is limited. In order to examine this usefulness, we have built two other types of models that implement the decomposition of layers into product of sparse matrices, either with random sparsity support, or with random weights, rather than considering those provided by `Palm4MSA`:

- "PSM random": We construct completely random sparse factorization. The sparsity support is chosen by projecting a reduced centered Gaussian matrix. The weights are initialized using the procedure described in Section 3.1;
- "PSM re-init.": We use the sparsity support obtained by `Palm4MSA` but we re-initialize the weights using the procedure described in Section 3.1.

Table 1 shows the network with the `Palm4MSA` method obtains the best performance in classification after refining the network (more results in Table 4 of Supplementary Material A.7). Overall «PSM re-init» and «PSM random» may perform very badly on some cases, which suggests the importance of both weights and sparsity support found by `Palm4MSA`.

| | MNIST Lenet | SVHN VGG19 | CIFAR10 VGG19 | CIFAR100 VGG19 | CIFAR100 Resnet20 | CIFAR100 Resnet50 |
|---|---|---|---|---|---|---|
| PSM Q=3 K=14 | 0.99 | 0.95 | 0.92 | 0.62 | 0.70 | 0.72 |
| PSM re-init. Q=3 K=14 | 0.99 | 0.89 | 0.84 | 0.31 | 0.60 | 0.58 |
| PSM random Q=3 K=14 | 0.99 | 0.93 | 0.84 | 0.51 | 0.60 | 0.59 |

Table 1: Performance of NN models compressed by 3 techniques of layer decomposition into sparse matrix products. "PSM" refers to the proposed procedure using `Palm4MSA` on the pre-trained layers; "PSM re-init" uses `Palm4MSA` to define the sparsity support but the weights are re-initialized; "PSM random" randomly initializes weights and sparsity support.

**Sparsity level and the number of factors.** Sparsity level corresponds to the approximate number of non-zero values in each row and each column of the sparse matrices constituting the sparse decomposition. Figure 1 and Table 4 show the performance of our model

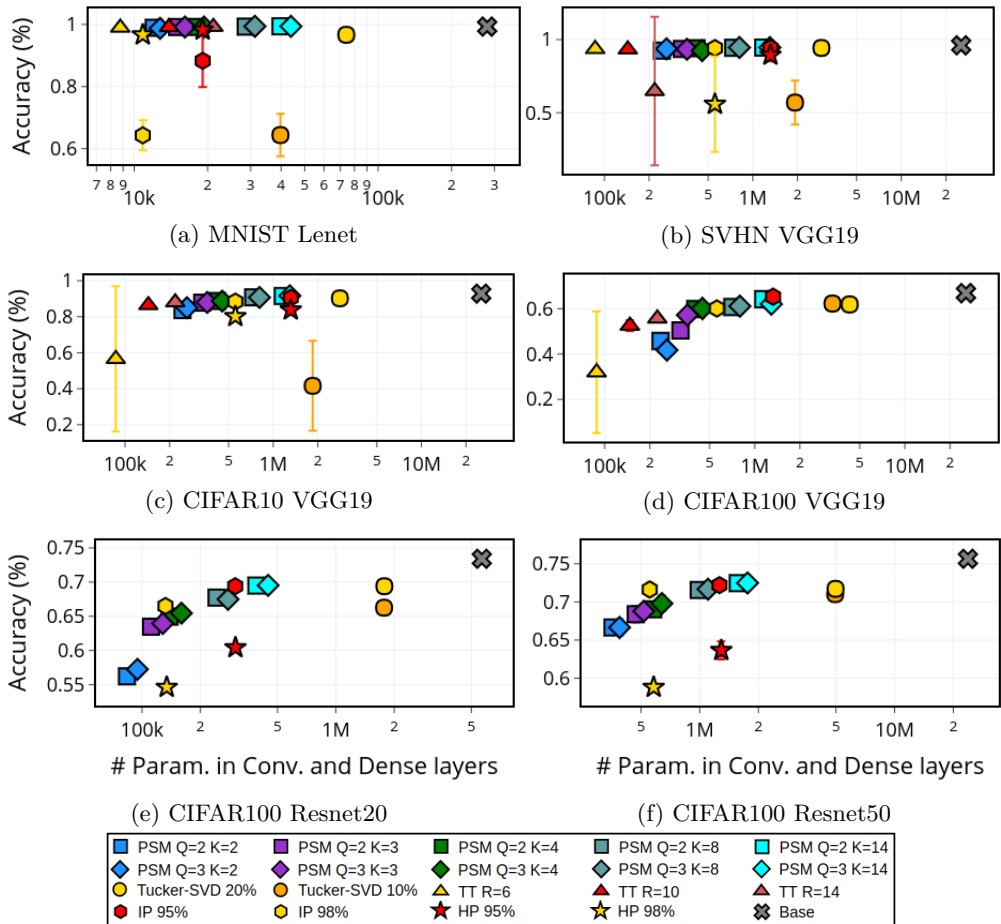

Figure 1: Accuracy vs number of parameters (log scale) for compressing standard architecture (pretrained models) on few datasets. Comparative results of our approach (*PSM*), the hybrid SVD-Tucker (*Tucker-SVD*), the TT method (*TT*), the Hard (*Hard Pruning*) and iterative (*Iterative Pruning*) pruning techniques. Details on the hyperparameters for each tested method are included in the name of the methods. *Base* stands for the uncompressed pretrained model.

obtained with various sparsity levels $K$ and numbers of factors $Q$. We notice that the number of factors seems to have a rather limited effect on the quality of the performance, while sparsity level is a more determining factor of the quality of the approximation. Note that we did not considered the Hierarchical version of `Palm4MSA` (Magoarou & Gribonval, 2016) since it requires $Q = \log D$ factors, which greatly increases the number of non-zero values in the decomposition without significantly improving the final performance of the model.

### 3.3 COMPARATIVE STUDY

Figure 1 reports comparative results gained on standard benchmark datasets with well known architectures. We discuss below the main comments to be drawn from these results.

**Reliability.** First of all, the behaviour of the baseline methods seems quite dependent on the experiment. For instance *TT* performance may vary depending on the chosen rank (e.g. see rank 10 in figure 1-(b)); *Hard Pruning* technique performs badly on MNIST. Moreover these baselines are not always manageable in practice (e.g. no results of *TT* on Resnet compression, see below). On the contrary, we observe some more stable performances with

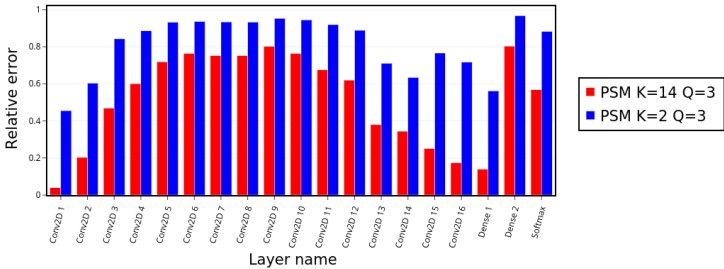

Figure 2: Relative error approximation by layers using `Palm4MSA` for `VGG19` architecture.

regards to the choice of hyper-parameters and a systematic very low variance obtained with our method.

**Comparison to low rank decomposition techniques.** Our approach significantly outperforms the *Tucker-SVD* method in any case. We believe that the low rank approximation may make more difficult to reach high compression rates while preserving good performance. This emphasizes our point that standard low rank decomposition methods cannot afford high compression regime, *e.g.* low-rank assumption, without degradation of the accuracy of the network. On the contrary, the *TT* formulation may achieve higher compression rates than *Tucker*, as was already observed in past works. It seems better suited than Tucker decomposition and may performs similarly or better as our method in some cases. Yet, the method has few drawbacks: first it may exhibit very strong variance, especially for high compression rates (see results in figures 1-(b) to 1-(d) on `SVHN`, `CIFAR10-100`) ; second, as illustrated in Supplementary Material A.6, the implementation provided by authors do not allow getting results in any case, when the product of the number of filters and of the *TT* rank is large. In particular we were unable to run experiments on models such as `Resnet20` and `Resnet50` because the memory footprint is increased considerably (figures 1-(e) and 1-(f)). We thus are unable to get results for higher compression rates that those in the figure with VGG19 (figures 1-(c) and 1-(d)).

**Comparison with pruning techniques.** In the "Hard" pruning case, the compressed network perform very badly. This confirms that a sparse factorization with more than one factor is profitable. When applying the procedure of Zhu & Gupta (2017), however, the magnitude based pruning method preserve good accuracy while removing up to 98% of the connections from the model, except for the `MNIST` dataset. While our approach significantly outperforms the *Hard pruning* technique in any case, its *Iterative pruning* variant Zhu & Gupta (2017) may sometimes leads to significantly higher performance compression in high compression settings than our approach, this is the case in particular with Resnet models on `CIFAR100` (figures 1-(e) and 1-(f)). Otherwise, in other settings on Resnet models, and for compressing other models, this technique allows similar performance vs compression trade-off than our method. Since the *Hard pruning* technique may be viewed as a special case of our method, this suggests that an iterative-like extension of our method could reach even better results, which is a perspective of this work.

## 4 CONCLUSION

The proposed approach is able to compress dense and convolutional layers of a neural network by decomposing weight matrices into sparse matrices products. Unlike common decomposition methods, our method does not make any assumptions on the rank of the weight matrices and allows high compression rate while maintaining good accuracy. The experiments show the interest of using a product of multiple sparse matrices instead of a single sparse matrix to convey, in theory, more information for a fixed number of non-zero values. A possible extension of this work would be to study a strategy for progressive sparcity inducing Zhu & Gupta (2017) that could offer further improvements.

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

## A    SUPPLEMENTARY MATERIAL

### A.1    ILLUSTRATION OF THE PROPOSED METHOD

The proposed method allows network compression through the factorization of layers weights. Figure 3 highlight the difference between the standard and compressed architecture composed of a convolutional and a fully connected layer, where each layer weight matrix is replaced by a product of sparce matrices.

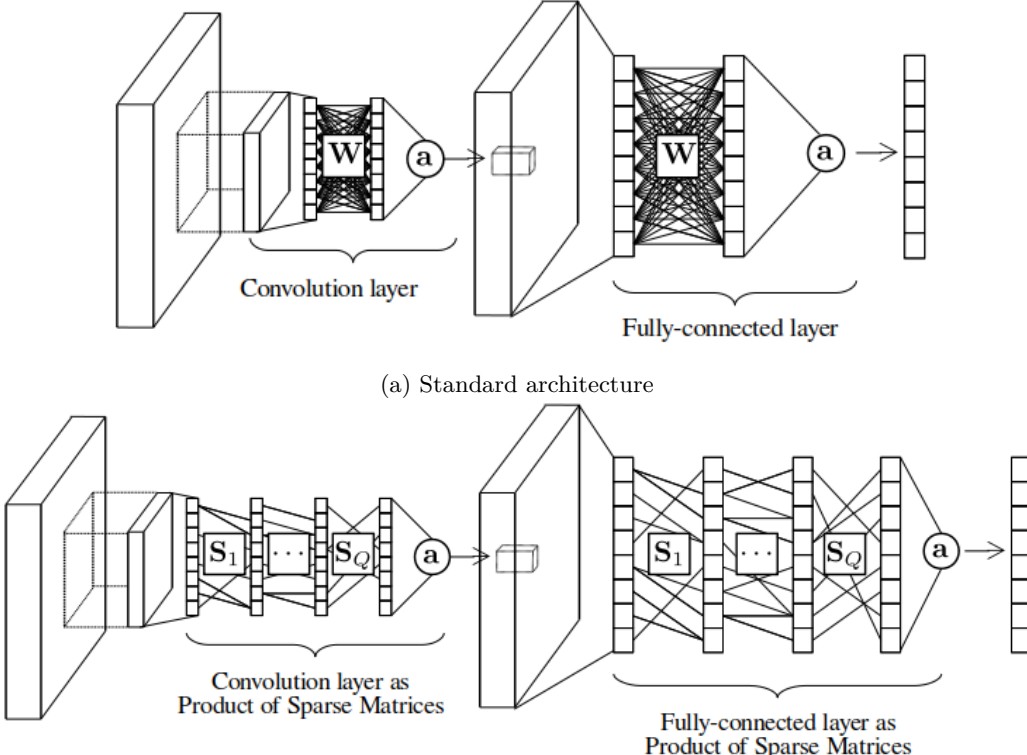

(a) Standard architecture

(b) Compressed architecture

Figure 3: Illustration of the proposed sparse factorization method.

### A.2    RESHAPE OPERATIONS IN CONVOLUTIONAL LAYERS

In order to apply factorization of convolutions weights, we employ reshaping operators to represent the linear operator as a dense matrix applied to all the patches extracted from an input (Garipov et al., 2016), see Figure 4 for more details.

### A.3    IMPLEMENTATION DETAILS

IMPLEMENTATION OF SPARSE MATRICES.    In our implementation, a sparse matrix is in fact defined as the Hadamard product $\odot$, or pairwise product, between a dense matrix that encodes the weights to be learned and a matrix of constant binary elements that encodes the sparsity support. Thus, the implementation of a sparse matrix $\mathbf{S} \in \mathbb{R}^{D \times D}$, $\|\mathbf{S}\|_0 = \mathcal{O}(D)$ is:

$$\mathbf{S} = \mathbf{W} \odot \mathbf{M}, \qquad (4)$$

where $\mathbf{W} \in \mathbb{R}^{D \times D}$ is a dense weight matrix and $\mathbf{M} \in \{0,1\}^{D \times D}$, $\|\mathbf{M}\|_0 = \mathcal{O}(D)$ is a constant binary matrix. With this implementation, the values of the gradient weights of $\mathbf{W}$

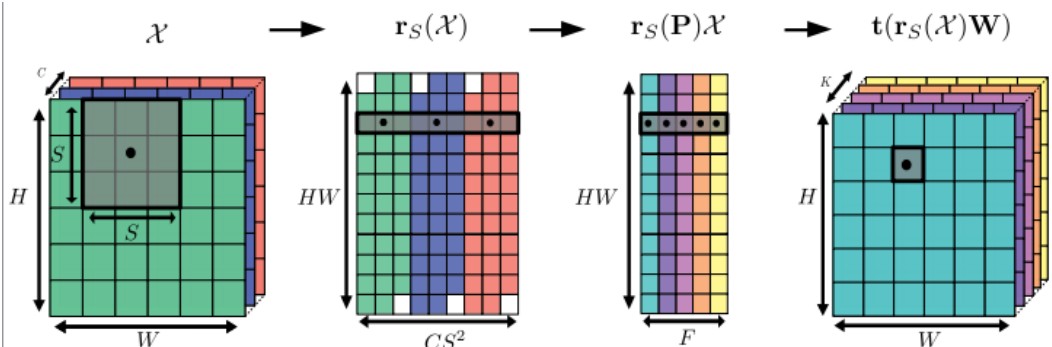

Figure 4: Description of the reshape operations and locally linear transformations computed in the convolutional layers. The grey box represent the receptive field of the convolution filters at the black dot coordinate. The white squares in the second step correspond to the zero padding of the input.

outside the sparsity support defined by $\mathbf{M}$ are always equal to zero and thus the corresponding weights of $\mathbf{W}$ are never updated. In other words, the sparsity support of $\mathbf{S}$ is fixed by $\mathbf{M}$. Although this implementation allows to evaluate ou method, the dense storage of all the values of $\mathbf{W}$ and $\mathbf{M}$ is required. Specifically, $2D^2$ values are stored to simulate the sparsity of a matrix of size $D^2$ containing $\mathcal{O}(D)$ non-zero values. This non-optimal implementation takes advantage of the parallel algorithms of the matrix product and is actually faster on GPU than an implementation that would use Tensorflow's `SparseTensor` class [1].

IMPLEMENTATION OF THE CONVOLUTION. To compute the convolutions in the network, we rebuild the convolution kernel from the product of sparse matrices. Then this convolution kernel is directly used as a parameter of the Tensorflow function `conv2d` for fast computation.

IMPLEMENTATION OF THE TENSORTRAIN DECOMPOSITION. The decomposition is performed by applying the decomposition function `matrix_product_state` provided by the Tensorly library on the tensors obtained on the pre-trained networks.

IMPLEMENTATION OF PRUNING AS A FUNCTION OF MAGNITUDE. To implement the method from Zhu & Gupta (2017), we used the function `prune_low_magnitude` from the library `tensorflow_model_optimization` provided by Tensorflow. With this method, the pruning and the refinement of the weights are combined by progressively removing the connections during the learning process until the desired percentage of pruning is obtained.

(RE)-INITIALIZATION OF THE WEIGHTS OF A SPARSE MATRIX DECOMPOSITION. When the weights of a sparse matrix factorization are not provided by `palm4MSA`, the initialization of the weights must be adapted to the reduced number of connections in the layer. We adapt the `Xavier` (Glorot & Bengio, 2010) and `He` (He et al., 2015) initialization methods for sparse matrices. Specifically, the first sparse factor is initialized using the `He` method because *ReLU* activation function is apllied yielding values that are not zero-centered. The following sparse factors are initialized using the `Xavier` method since there is no non-linearity between factors.

### A.4 `palm4MSA` ALGORITHM

The `palm4MSA` algorithm Magoarou & Gribonval (2016) is given in Algorithm 1 together with the time complexity of each line, using $A = \min(D, D')$ and $B = \max(D, D')$ for a matrix to factorize $\mathbf{W} \in \mathbb{R}^{D \times D'}$. Even more general constraints can be used, the constraint

---

[1]One can use the `SparseTensor` in conjunction with the `Variable` class to implement these layers sparingly and have exactly a reduced number of parameters but this implementation was slower and we preferred not to use it for experiments.

sets $\mathcal{E}_q$ are typically defined as the intersection of the set of unit Froebenius-norm matrices and of a set of sparse matrices. The unit Froebenius norm is used together with the $\lambda$ factor to avoid a scaling indeterminacy. Note that to simplify the model presentation, factor $\lambda$ is used internally in `palm4MSA` and is integrated in factor $\mathbf{S}_1$ at the end of the algorithm (Line 14) so that $\mathbf{S}_1$ does not satisfy the unit Froebenius norm in $\mathcal{E}_1$ at the end of the algorithm. The sparse constraints we used, as in Magoarou & Gribonval (2016), consist of trying to have a given number of non-zero coefficients in each row and in each column. This number of non-zero coefficients is called sparsity level in this paper. In practice, the projection function at Line 9 keeps the largest non-zero coefficients in each row and in each column, which only guarantees the actual number of non-zero coefficients is at least equal to the sparsity level.

---

**Algorithm 1** `palm4MSA` algorithm

---

**Require:** The matrix to factorize $\mathbf{U} \in \mathbb{R}^{D \times D'}$, the desired number of factors $Q$, the constraint sets $\mathcal{E}_q$ , $q \in [\![Q]\!]$ and a stopping criterion (e.g., here, a number of iterations $I$ ).
1: $\lambda \leftarrow ||S_1||_F$ $\{\mathcal{O}(B)\}$
2: $S_1 \leftarrow \frac{1}{\lambda}S_1$ $\{\mathcal{O}(B)\}$
3: **for** $i \in [\![I]\!]$ while the stopping criterion is not met **do**
4:     **for** $q = Q$ down to 1 **do**
5:         $\mathbf{L}_q \leftarrow \prod_{l=1}^{q-1} \mathbf{S}_l^{(i)}$
6:         $\mathbf{R}_q \leftarrow \prod_{l=q+1}^{Q} \mathbf{S}_l^{(i+1)}$
7:         Choose $c > \lambda^2 ||\mathbf{R}_q||_2^2 ||\mathbf{L}_q||_2^2$ $\{\mathcal{O}(A \log B + B)\}$
8:         $\mathbf{D} \leftarrow \mathbf{S}_q^i - \frac{1}{c}\lambda\mathbf{L}_q^T \left(\lambda\mathbf{L}_q\mathbf{S}_q^i\mathbf{R}_q - \mathbf{U}\right)\mathbf{R}_q^T$ $\{\mathcal{O}(AB \log B)\}$
9:         $\mathbf{S}_q^{(i+1)} \leftarrow P_{\mathcal{E}_q}(\mathbf{D})$ $\{\mathcal{O}(A^2 \log A) \text{ or } \mathcal{O}(AB \log B)\}$
10:     **end for**
11:     $\hat{\mathbf{U}} := \prod_{j=1}^{Q} \mathbf{S}_q^{(i+1)}$ $\{\mathcal{O}(A^2 \log B + AB)\}$
12:     $\lambda \leftarrow \frac{Trace(\mathbf{U}^T\hat{\mathbf{U}})}{Trace(\hat{\mathbf{U}}^T\hat{\mathbf{U}})}$ $\{\mathcal{O}(AB)\}$
13: **end for**
14: $S_1 \leftarrow \lambda S_1$ $\{\mathcal{O}(B)\}$
**Ensure:** $\{\mathbf{S}_q : \mathbf{S}_q \in \mathcal{E}_q\}_{q \in [\![Q]\!]}$ such that $\prod_{q \in [\![Q]\!]} \mathbf{S}_q \approx \mathbf{U}$

---

## A.5 DATASET DETAILS

Experiments are conducted on four public image classification dataset: MNIST, SVHN, CIFAR10, and CIFAR100, with three pretrained network architectures: Lenet, VGG-19, Resnet50, and Resnet20. Table 2 details the datasets' characteristics and the corresponding NN models on which we evaluated compression methods.

| Nom | Input shape | # classes | Train size | Validation size | Test size | NN models |
|---|---|---|---|---|---|---|
| MNIST | $(28 \times 28 \times 1)$ | 10 | 40 000 | 10 000 | 10 000 | Lenet |
| SVHN | $(32 \times 32 \times 3)$ | 10 | 63 257 | 10 000 | 26 032 | VGG19 |
| CIFAR10 | $(32 \times 32 \times 3)$ | 10 | 50 000 | 10 000 | 10 000 | VGG19 |
| CIFAR100 | $(32 \times 32 \times 3)$ | 100 | 50 000 | 10 000 | 10 000 | VGG19, Resnet50, Resnet20 |

Table 2: Datasets: attributes and investigated NN models.

## A.6 EXPERIMENTS ON TENSOR TRAIN MEMORY OVERHEAD

Although Tensor Train can obtain impressive compression rates the method may require large amounts of memory. This memory requirement does not allow to experiment on architectures with many convolution layers with many filters, such as ResNet. Table 3 highlights the increase in memory when experimenting on investigated architectures for few hyperparameter settings. The row *Others* stands for requirement of all other methods.

|  | MNIST Lenet | SVHN VGG19 | Cifar10 VGG19 | Cifar100 VGG19 | Cifar100 Resnet20 | Cifar100 Resnet50 |
|---|---|---|---|---|---|---|
| Others | 1024 | 65,536 | 65,536 | 65,536 | 65,536 | 262,144 |
| Tensortrain R=6 K=4 | 2,304 | 393,216 | 393,216 | 393,216 | 393,216 | 1,572,864 |
| Tensortrain R=10 K=4 | 3,840 | 655,360 | 655,360 | 655,360 | 655,360 | 2,621,440 |
| Tensortrain R=14 K=4 | 5,376 | 917,504 | 917,504 | 917,504 | 917,504 | 3,670,016 |

Table 3: Magnification of a sample's representation induced by the `Tensortrain` layers. The maximum number of non-zero values in a sampl's representatio during architecture execution is displayed.

## A.7 EXPERIMENTS OF THE PROPOSED METHOD

Since the error of the matrix approximation can be high, despite the relative good accuracy, we investigate other factorization method than `Palm4MSA`. Specifically, two other methods are evaluated. «PSM re-init.» use the same sparsity support as palm with re-initialized weights and «PSM random» has random sparsity support and weights. Table 4 present the results and shows the supperiority of `Palm4MSA`.

|  | MNIST Lenet | SVHN Vgg19 | Cifar10 Vgg19 | Cifar100 Vgg19 | Cifar100 Resnet20 | Cifar100 Resnet50 |
|---|---|---|---|---|---|---|
| Base | **0.99** | **0.96** | **0.93** | **0.67** | **0.73** | **0.76** |
| PSM Q=2 K=2 | 0.99 | 0.92 | 0.84 | 0.46 | 0.56 | 0.67 |
| PSM re-init. Q=2 K=2 | 0.99 | 0.82 | 0.81 | 0.42 | 0.53 | 0.57 |
| PSM random Q=2 K=2 | 0.98 | 0.91 | 0.81 | 0.44 | 0.48 | 0.41 |
| PSM Q=2 K=14 | 0.99 | 0.95 | 0.92 | 0.64 | 0.69 | 0.72 |
| PSM re-init. Q=2 K=14 | 0.99 | 0.44 | 0.86 | 0.57 | 0.63 | 0.63 |
| PSM random Q=2 K=14 | 0.99 | 0.92 | 0.85 | 0.58 | 0.62 | 0.62 |
| PSM Q=3 K=2 | 0.99 | 0.94 | 0.85 | 0.42 | 0.57 | 0.67 |
| PSM re-init. Q=3 K=2 | 0.98 | 0.91 | 0.80 | 0.32 | 0.48 | 0.51 |
| PSM random Q=3 K=2 | 0.98 | 0.90 | 0.79 | 0.39 | 0.29 | 0.47 |
| PSM Q=3 K=14 | 0.99 | 0.95 | 0.92 | 0.62 | 0.70 | 0.72 |
| PSM re-init. Q=3 K=14 | 0.99 | 0.89 | 0.84 | 0.31 | 0.60 | 0.58 |
| PSM random Q=3 K=14 | 0.99 | 0.93 | 0.84 | 0.51 | 0.60 | 0.59 |

Table 4: Performance of neural network models compressed by 3 techniques of layer decomposition into sparse matrix products. «PSM» refers to the proposed procedure using `Palm4MSA` on the pre-trained layers; «PSM re-init» uses `Palm4MSA` to define the sparsity support but the weights are re-initialized; «PSM random» randomly initializes weights and sparsity support.

