# OpenReview forum: "Sparse matrix products for neural network compression"
_ICLR.cc/2021/Conference — Reject_

### Official Review · AnonReviewer4 · 2020-10-23
**Reasonable, but not exciting, experimental study of known matrix compression ideas**

**Rating:** 4
**Confidence:** 3

**Review:**

The paper considers compression of neural network weight matrices by decomposition into a product of sparse matrices. Its main contribution is an experimental study of the effectiveness of this type of compression in neural networks for image classification (MNIST and CIFAR) with standard architectures such as RESNET and VGG19. The compression algorithm, Palm4MSA, is taken from previous work. I am not familiar with other experimental studies of Palm4MSA, but the paper has a good overview of other compression methods that I am familiar with, so I trust that the overview is complete.

The main reason that I am not too excited about the paper is that the results are kind of expected. We know that architectures with much smaller number of parameters exist (a comparison with MobileNetV2 which has about 40x fewer parameters would be suitable).

As a side remark, I don't think it makes a lot of sense to distinguish between neural architecture search and neural network compression. You can think of neural architecture search as yet another compression method that happens during training (not post training).

One way of making the paper more interesting would be to consider the compression objective already during training, rather than merely fine tuning post compression. The "PSM random" and "PSM re-init" methods seem like rather arbitrary choices for comparison.

---

> ### Author Response · Authors · 2020-11-23
> **We would like to thank the reviewer for their review.**
>
> We understand the reviewer's concerns about our experiment section but we would like to emphasize the idea that our method is not to beat the state of the art approaches at all cost. We rather propose a general and flexible idea, that is to express layer weight matrices as products of sparse matrices. We also proposed a very simple procedure to implement this idea but it is clear that there are possible improvements in order to beat existing methods. At the moment, we essentially illustrate that our approach to model compression is able to reach results close to the state of the art. In this work, we show that PSM layers are serious contenders to achieve model compression. This same idea motivates the comparison with 'PSM random' and 'PSM re-init' methods. Indeed, these experiments show that the weights and sparse support returned by Palm4MSA are of crucial interest for the network to be able to learn something. In that case, the natural question that arises is whether or not other strategies than Palm4MSA (or improvements of it) can be used to build PSM layers that achieve better performance. We believe it is the case and we hope the community can use our work as a basis to find these better compression strategies.
> Finally, evaluating the compression of light models such as MobileNet is quite interesting and should be included in further experiments.

---

### Official Review · AnonReviewer3 · 2020-10-28
**Currently recommending rejection**

**Rating:** 4
**Confidence:** 5

**Review:**

**Summary**: The paper proposes compressing the layers of the neural networks using a product of sparse matrices. This approach is in line with the initial methods on neural network compression: direct (task-independent) compression of weights, which is followed by NN task-dependent fine-tuning. In this case, the direct compression is obtained using the Palm4MSA method of Magoarou and Gribonval (2016), and then models are fine-tuned in an end-to-end fashion using TensorFlow.

**Decision**: Given the contributions of the paper and the lackluster experimental evaluation, I am recommending rejection.

**Detailed comments**:
The factorization as a product of sparse matrices is an interesting approach, and in its general form, such a scheme has not been studied in the context of neural network compression. However, the evaluation of the proposed approach is not convincing:

 - Effect of chaining sparse matrices for the actual inference is not discussed. As authors are well aware, the support of sparse matrix-vector products is limited in major frameworks and hardware. Therefore, chaining multiple sparse products might significantly delay the actual inference, even though it has fewer parameters and theoretically fewer FLOPs.

 - Comparison of different compression mechanism for neural networks is a very complex task: since most methods are applied per layer, the parameters of the compressions must be tuned per layer too: e.g., ranks/sparsity of every layer cannot have the same value, as some layers require more/fewer parameters. However, in the comparison, authors use uniform settings across layers for their method (e.g., M=4 cores) and for other methods. The conclusions under such comparison strategy have a limited value: it says that chosen hyperparameters of compression A beats the chosen (by hand) hyperparameters of compression B in terms of final compression ratio/tradeoff etc. It leaves the possibility that by tuning parameters of B we might outperform A. To fully understand the usefulness of the compression scheme, we need to ask a different question: given the scheme A and B (with the best selection of hyperparameters for both), compression using which scheme gives the smallest model for a given accuracy?  Or even better, which compression results in a better tradeoff when optimizing the shape/form of the schemes to suit the compression target?

- Related to the previous point, authors miss a large body of the methods that instead of expecting the weights to be in a certain form (e.g., sparse or low-rank), actually force the network to attain such form using penalties and constraints. For example, in terms of low-rank compression, here are some of the relevant works:
   1. Accelerating Very Deep Convolutional Networks for Classification and Detection (IEEE TPAMI 2016)
   2. Coordinating Filters for Faster Deep Neural Networks (ICCV 2017)
   3. Compression-Aware Training of Deep Networks (NIPS2017)
   4. Constrained Optimization Based Low-rank Approximation of Deep Neural Networks (ECCV 2018)
   5. Automated Multi-Stage Compression of Neural Networks (ICCV Workshops 2019)
   6. Low-Rank Compression of Neural Nets: Learning the Rank of Each Layer (CVPR 2020)
   7. Factorized Higher-Order CNNs with an Application to Spatio-Temporal Emotion Estimation (CVPR 2020)
   8. TRP: Trained Rank Pruning for Efficient Deep Neural Networks (IJCAI 2020)

There is an equally large body of literature on network sparsification. Please add the comparison to best-in-its-class results from the literature in order to fully evaluate the proposed scheme.

*Minor concerns*:
I strongly believe citing Li Deng for the MNIST dataset is inadequate.  Please correct.

*Post rebuttal comments*: I appreciate the author's efforts for the rebuttal, however, the feedback did not adequately address my questions. I am not changing the score.

---

> ### Author Response · Authors · 2020-11-23
> **We would like to thank the reviewer for their review and the provided references.**
>
> - We will provide a similar answer to AnonReviewer2, the first reviewer on this page. We can’t deny that, in the current state of available hardware, an architecture with PSM layers could not compete with usual dense architectures, especially if run on modern GPU where most multiplications can be run simultaneously. However, we think that the accumulation of papers achieving interesting performances with sparsity based techniques can provide incentives to develop new hardware that would benefit from sparse multiplications.
> - In fact, concerning the Tucker decomposition, we use the VBMF strategy to discover the ideal rank for each layer; concerning the iterative sparsity inducing technique, we use a global compression rate for the network but no particular constraint is set so that every layer should have the same compression rate: at each iteration a percentage of the global network weights are removed. The reviewer is correct concerning the other compression methods (tensortrain and PSM) and the general idea that it would be better to tune every compression strategy for every layer to achieve the best possible performance with a given compression rate. However, the combinatory of such experiment is so huge that only a few laboratories in the world could conduct it. And even they would require to make some arbitrary choices at some point. We do not have the infrastructure for such experiments so we chose the approach that seemed the fairest: our method is actually at a disadvantage compared to Tucker and Iterative Sparsity and tensortrain induces a huge overhead in convolution layers, which makes it unusable for wide networks. Anyway, we do not pretend that our technique is so good that it would beat every existing compression scheme. We believe that the idea of defining weight matrices as product of sparse matrices is very flexible and could set the basis for a new line of research for compressing neural networks. Our approach is simple yet it achieves good results, we think that can be inspiring for the community.
> - We particularly thank the reviewer to point out these works that are very interesting. Unfortunately, we doubt that such approach could be used for our method as the theory behind what makes a matrix a good candidate for being "sparse factorized" is not known yet. In the presence of such condition, it would be very interesting indeed to try to set constraint on the initial network to achieve this goal.

---

> > ### Comment · AnonReviewer4 · 2020-11-23
> > **Factorization as part of learning**
> >
> > You write: "Unfortunately, we doubt that such approach could be used for our method as the theory behind what makes a matrix a good candidate for being "sparse factorized" is not known yet."
> >
> > I think you are missing the point. If you in advance know that you want to end up with a model with a sparse factorization, it seems like a good idea to include this criterion in the learning algorithms, rather than treating it as a post-processing step. So instead of expanding a layer into two sparse layers, you can start with two layers and learn a sparse representation that you can prune as a last step. It is not clear that one needs a 'criterion for being a good candidate for being "sparse factorized"', as long as you make sure to learn models that are, e.g. through appropriate regularization.

---

> > > ### Author Response · Authors · 2020-11-24
> > > **Understood**
> > >
> > > Ok, thank you for the clarification.  I can think of a pretty simple strategy to achieve this. It would be interesting if it worked!

---

### Official Review · AnonReviewer1 · 2020-10-29
**a neural network compressing method, based on factorization of weight matrix to the products of multiple sparse matrices**

**Rating:** 5
**Confidence:** 3

**Review:**

The authors introduced a neural network compressing method, based on factorization of weight matrix to the products of multiple sparse matrices. The goal is to achieve high compression rate. The author used a previous algorithm (Palm4MSA) to implement the method. The experiment result is better than other low-rank-based method, but is similar or worse to Iterative pruning and TT method.

Pros:

- The introduced method is easy to understand and seems to make sense.
- The idea of using products of multiple sparse matrices to represent the weight matrix is nice.

Cons:

- The introduced method is almost direct application of existing algorithm (Palm4MSA).
- The experiment result is not state-of-the-art: it is worse than Iterative pruning. The authors stated that "an iterative-like extension of our method could reach even better results",  so it would be important to include these results in the paper.

Questions and suggestions:

- In Eq. 1, the \prod S_i x is confusing. (\prod S_i) x could be better.

- In Fig. 1, you can use compression rate rather than actual # of parameters as the measurement. You can also use a curve to better illustrate the compression-accuracy tradeoff.

---

> ### Author Response · Authors · 2020-11-23
> **We would like to thank the reviewer for their review.**
>
> When reading the "Cons" section of the reviewer, we realize that our message was not clear enough on the ambition of the paper. We believe that our work could pioneer a new approach to neural network compression, that is: simply expressing weight matrices as products of sparse matrices instead of simple dense matrices. This idea is agnostic of the method used for the compression phase of the network. In our case, we use the existing Palm4MSA algorithm as a proof of concept of the framework we propose but other methods may offer improvements on this part. We think that our work could set the basis for a line of research that would focus on finding other compression strategies leading to a compressed neural network with PSM layers.

---

### Official Review · AnonReviewer2 · 2020-10-31
**A solid work with some concerns**

**Rating:** 7
**Confidence:** 4

**Review:**

In this paper, the authors propose to impose sparsity upon the low-rank compressions methods. Imposing sparsity is generally interesting and meaningful. The experimental results are appreciated.

There are some concerns that make this paper cannot be fully appreciated:
1. There are so many structured matrix schemes to compress neural networks. Now, the authors claim sparsity + low-rank can provide better accuracy-efficiency tradeoff.
    This is surely not enough.  We compress neural networks for mobile computing devices, while the "best" accuracy-efficiency tradeoff is not the final goal, unless you are theoretically drawing the boundaries.
    A more important question with the proposed scheme is as follows: 1). the current compression schemes in the literature are good enough in balancing accuracy-efficiency. Sparsity will surely contributed. 2). however, when the compressed model is downloaded onto a mobile device, will sparse neural networks easy to run on such platforms?
   This is a naturally question that needs an answer.  The best accuracy-efficiency tradeoff is not the final goal;  a high cost-performance is the objective to optimize. I mean you may add sparsity to achieve better accuracy-efficiency tradeoff, but we do not allow it to introducing much computation overhead of sparse computations.

2. I have to raise the principle of "simple and effective" is the best, for this kind of tasks.
    The previous schemes (circulant, low-rank, sparsity only, etc) are simple and shown to be effective. Now, your model is more sophisticated and the overhead is higher. Will this solution serve the purpose of compressing neural networks for mobile devices?
    I would hope the authors do not answer too quickly.  The purpose of achieving better accuracy-efficiency tradeoff is finally making deep learning be practically deployed onto mobile devices, while computing power, memory, bandwidth and energy are limited.

---

> ### Author Response · Authors · 2020-11-23
> **We would like to thank the reviewer for their positive review.**
>
> We understand the reviewer concerns regarding the possible deployment of our approach to mobile devices.
> It is true that we have in mind the possible benefits of using an architecture with PSM layers on mobile devices because sparse matrices product are of particular interest when parallelization is not possible in the absence of GPU hardware.
> However, we do not believe that our compression strategy could be used in production yet, because the overhead induced by sparse matrices products would nullify the gains in term of FLOP offered by PSM layers.
> We hope that the accumulation of papers involving sparsity with such compression capabilities would encourage the community to develop new hardware suited for sparse matrix multiplication, as the paper: "Hardware support for efficient sparse matrix vector multiplication".
> We believe that having dedicated hardware for sparse linear algebra would decrease FLOP and improve accuracy-efficiency trade-off.

---

### Decision · Program_Chairs · 2021-01-07
**Final Decision**

**Decision:**

Reject

**Comment:**

The idea of using multiple sparse matrices seems to be new, but the novelty of the idea alone isn't enough to convince the AC and reviewers (indeed, the idea might not be new, but has never been discussed in literature because of the drawbacks we discuss here). As the authors and reviewers/AC seem to agree, the actual benefits of sparse matrix multiplies are hard to realize, especially on embedded devices, so the contributions at this point are mainly hypothetical and only about the new idea. Each reviewer brought up issues (even the most positive reviewer) and mostly the reviewers were not persuaded by the rebuttal. In short, there wasn't evidence that this new idea could really contribute to the state-of-the-art.  This is now a fairly crowded topic (e.g., all the  papers brought up by R3 in just that one class of methods), and new papers should beat state-of-the-art and/or introduce new theory -- an example would be a paper from last year's ICLR, https://openreview.net/forum?id=HJfwJ2A5KX , "Data-Dependent Coresets for Compressing Neural Networks with Applications to Generalization Bounds" (Baykal, Liebenwein, Gilitschenski, Feldman, Ru) which not only gives an efficient technique (not based on sparsity of weights) but also gives types of generalization guarantees.

As R1 said, the results are not state-of-the-art, and we have to believe the authors that "an iterative-like extension of our method could reach even better results". The rebuttal says that the paper's goal is to "pioneer a new approach to neural network compression". But if you can get better results with something better than Palm4MSA, then please do so, and demonstrate the evidence!  Right now, the paper assumes we could implement sparse multiplication efficiently on embedded devices, and assumes we could get better results: both these are quite hypothetical. The AC encourages a resubmission of this paper after these results have been addressed.